# Musculoskeletal surgeons have implicit bias towards the biomedical paradigm of human illness

**Sina Ramtin, Dayal Rajagopalan, David Ring◉\*, Tom Crijns, Prakash Jayakumar**

The University of Texas Dell Medical School, Austin, Texas, United States of America

\* David.Ring@austin.utexas.edu

**Data Availability Statement:** All relevant data are within the manuscript and its Supporting information files.

**Funding:** The author(s) received no specific funding for this work.

## Abstract

### Background

Evidence is mounting that the biopsychosocial paradigm is more accurate and useful than the biomedical paradigm of care. Habits of thought can hinder the implementation of this knowledge into daily care strategies. To understand and lessen these potential barriers, we asked: 1) What is the relative implicit and explicit attitudes of musculoskeletal surgeons towards the biomedical or biopsychosocial paradigms of medicine? 2) What surgeon factors are associated with these attitudes?

### Methods

An online survey-based experiment was distributed to members of the Science of Variation Group (SOVG) with a total of 163 respondents. Implicit bias towards the biomedical or biopsychosocial paradigms was measured using an Implicit Association Test (IAT) designed by our team using open-source software; explicit preferences were measured using ordinal scales.

### Results

On average, surgeons demonstrated a moderate implicit bias towards the biomedical paradigm (d-score: -0.21; Interquartile range [IQR]: -0.56 to 0.19) and a moderate explicit preference towards the biopsychosocial paradigm (mean: 14; standard deviation: 14). A greater implicit bias towards the biomedical paradigm was associated with male surgeons (d-score: -0.30; IQR: -0.57 to 0.14; $P$ = 0.005). A greater explicit preference towards the biomedical paradigm was independently associated with a European practice location (Regression coefficient: -9.1; 95% CI: -14 to -4.4; $P$ <0.001) and trauma subspecialty (RC: -6.2; 95% CI: -11 to -1.0; $P$ <0.001).

### Conclusions

The observation that surgeons have an implicit bias favoring the biomedical paradigm might inform strategies for implementation of care strategies based on evidence favoring the biopsychosocial paradigm.

**Competing interests:** NO authors have competing interests pertinent to this study.

## Introduction

The biopsychosocial model posits that human illness results from the interplay of pathophysiology with thoughts, emotions, and circumstances [1–5]. The biomedical model of human illness considers symptoms as directly corresponding to underlying pathology [6, 7]. The association between variation in mental health (e.g., unhelpful thoughts and unhelpful feelings) and social well-being (e.g., security in life roles, connectedness, housing, food, job, and finances) with variation in pain intensity and magnitude of incapability [5, 8–12] relatively independent of pathology severity supports the utility of the biopsychosocial model [12–16]. Musculoskeletal specialists increasingly incorporate elements of the biopsychosocial model into daily practice [17], providing whole-person care that considers emotional needs and social circumstances, as well as the physical condition, while limiting low value tests and treatments [18]. Implementation of this line of evidence might be hindered by ingrained habits of thought and default actions (implicit bias) towards the biopsychosocial model.

Musculoskeletal specialists are well-positioned to discern the relative correspondence between pathophysiology and magnitude of incapability. There is compelling evidence that relatively greater magnitude of incapability signals mental health opportunities such as unhelpful thinking and feelings of worry or despair, along with social health opportunities such as insecurity in roles, finances, housing, and food [19, 20]. Surgeon implicit bias contrary to this evidence could harm patients through missed opportunities to address mental and social health in combination with exposure to tests and treatments with more potential for harm than benefit.

Prior research suggests the presence of discordance between implicit and explicit attitudes for controversial or difficult subjects [21]. Presumably no surgeon would say they don't appreciate the importance of mental and social factors on their patients' overall health. Implicit and explicit attitudes tend to use different modes of cognition (e.g., automatic/intuitive vs. critical/analytical) [21]. We might think of the explicit attitude as the aspiration and the implicit attitude as the starting point, with the difference between the two being the gap to be bridged to improve whole-person, comprehensive care.

Implicit attitude tests (IAT) can measure unconscious habits of thought such as a tendency to link obesity with laziness [22–26], or being a surgeon with being male [27]. Based on experience developing IATs to test implicit biases, we developed one to measure the implicit bias of musculoskeletal surgeons towards either the biomedical or biopsychosocial model of illness. Awareness of surgeon implicit bias towards the biomedical model might help promote debiasing strategies that support consideration of mental and social health in musculoskeletal care.

Therefore, we asked: What is the average relative implicit bias of musculoskeletal surgeons towards health as biomedical or biopsychosocial? 2) What is their relative explicit preference? 3) Which surgeon factors are associated with greater explicit preference or implicit bias towards either the biomedical or biopsychosocial paradigms? 4) Is there a correlation between the explicit preference and the implicit bias towards either paradigm?

## Materials and methods

### Study design and setting

We obtained approval for this cross-sectional study from our Institutional Review Board. We created an online survey-based experiment (Qualtrics, Provo, Utah, USA) and distributed the survey among all members of the Science of Variation Group (SOVG) between February 1st and March 1st, 2021. The introduction to the survey stated explicitly that completing the survey represented informed consent to use their anonymized data. No form of written or verbal

consent was required by the institutional review board. The SOVG is an international consortium of musculoskeletal surgeons (including general orthopedic surgeons, hand surgeons and other specialists, and fracture surgeons [often general surgeons in Europe]) who participate in monthly survey-based experiments to better understand variations in musculoskeletal care. The experiments conducted using the SOVG rely on within-sample variation to find associations. There is sufficient variation in survey responses among participants that the associations are likely reproducible. Take for instance the factors associated with implicit attitudes. The absolute rates, like mean scores for explicit and implicit attitudes, are likely not reproducible. The SOVG primarily consists of orthopedic surgeons who are white, male, academic, and based in the US or Europe. The respondent demographics are not by design but reflect the current lack of diversity in the field. We welcome new members and collaborators (https://www. surveymonkey.com/r/SOVG_FB). Our survey-based experiment was written in English and consisted of two components designed to 1) measure implicit biases using an implicit association test, and 2) to measure explicit attitudes and preferences for either the biomedical or biopsychosocial model on slider scales. We administered the explicit measure at the end to avoid priming respondents before doing the IAT.

There are about two hundred SOVG surgeons that participate in at least one survey experiment per year. One hundred and sixty-three surgeons completed this experiment. The majority were men (n = 141, 90%), and practice in the United States (n = 86, 53%) or Europe (n = 52, 32%). More than half of surgeons had 11 or more years of experience (n = 96, 61%), were fellowship-trained in trauma (n = 62, 40%) or hand surgery (n = 51, 33%) (Table 1). While the gender distribution of our study demographic is representative of the field of

**Table 1. Surgeon demographics.**

| Variable | Value |
|---|---|
| N | 163 |
| Gender | |
| Men | 141 (90%) |
| Women | 15 (9.6%) |
| Practice location | |
| United States | 86 (53%) |
| Europe | 52 (32%) |
| Other | 25 (15%) |
| Years in practice | |
| 0 to 5 | 29 (19%) |
| 6 to 10 | 31 (20%) |
| 11 to 20 | 51 (33%) |
| More than 20 | 45 (29%) |
| Subspecialty | |
| Hand and wrist | 51 (33%) |
| Trauma | 62 (40%) |
| Shoulder and elbow | 17 (11%) |
| General orthopedics/other | 26 (17%) |
| Supervising trainees | |
| Yes | 128 (82%) |
| No | 28 (18%) |

Discrete variables as number (percentage). Seven surgeons had missing data.

orthopedic surgery (around 90% men) it may not adequately represent associations of gender with implicit or explicit bias.

## Implicit Association Test (IAT)

An IAT is based on the premise that the longer it takes a respondent to associate a concept with positive descriptors, the more the concept is contrary to current habits of thought (implicit bias), and vice versa [28]. The IAT works by showing a concept on screen along with a descriptive category on each side (right and left). Someone taking the IAT can press a button on their keyboard that sorts the concept into either the right or the left category. The speed and accuracy with which respondents sort the concept into a category over a series of blocks is distilled into a single score, the d-score. The conventional d-score calculation involves dropping the first two trials in each testing block (blocks 4 and 7), recoding trial latencies outside of the established bounds (300 milliseconds to 3000 milliseconds) to being equivalent to the closest bound, log transforming the latency values before averaging them, including latency values from trials that were answered incorrectly, and finally omitting altogether data from respondents whose latencies are far higher than what is expected for the subject being analyzed [21].

The utility and validity of IATs has a strong basis in prior studies, and IATs have been used in a wide range of topics to measure implicit cognitive biases [29–31]. We created an IAT using IATgen to test musculoskeletal surgeons' bias toward the biomedical or biopsychosocial paradigms [32]. Through a working group discussion, we created a list of concept terms (words that are associated with either the biomedical or biopsychosocial paradigm) and attribute terms (words that have a positive or negative connotation) (Table 2). The authors selected terms we thought were a) representative of the concept we wanted to test and b) universal, so that experiment participants around the world would understand them. Terms were introduced by one team member and adjusted/approved by the others until consensus was reached. The working group consisted of multiple orthopedic surgeons and orthopedic research fellows. The IAT was conducted as outlined by Greenwald et al. and consisted of 7 association blocks. Blocks 1, 2, 3, 5, and 6 were for teaching, while blocks 4 and 7 were for evaluation (Table 3) [33]. Blocks 1, 2, 3, 5, and 6 were essentially "practice" blocks that were not scored and were in place to help respondents acclimate to the IAT itself before they completed blocks that were scored. The IAT is meant to test automatic or intuitive thinking, but this works best when respondents are fully acclimated to the format of the test. Participants are asked to sort the concept terms with the attribute terms as quickly as possible. During a testing block, we recorded the time needed to make an association to the millisecond. There are four

**Table 2. Concept and attribute terms.**

| Concept terms A: Biopsychosocial | Concept terms B: Biomedical | Attribute terms A: Positive | Attribute terms B: Negative |
|---|---|---|---|
| Mood | Physical exam | Positive | Negative |
| Attitude | Joint space | Meaningful | Inconsequential |
| Therapy | Range of motion | Constructive | Destructive |
| Coping strategies | MRI | Helpful | Ineffective |
| Mental health | Biomechanics | Important | Unimportant |
| Resiliency | Osteophyte | Excellent | Misfortune |
| Emotion | Arthroplasty | Trustworthy | Biased |
| Housing security | Stem cells | Vital | Horrible |
| Psychology | Arthroscopy | Life | Tragedy |
| Socioeconomic status | CT | Beneficial | Detrimental |

**Table 3. Implicit Association Test (IAT) structure.**

| Block* | Number of Trials | Round | Left-key response† | Right-key response† |
|---|---|---|---|---|
| 1 | 20 | Practice | Biopsychosocial | Biomedical |
| 2 | 20 | Practice | Positive | Negative |
| 3 | 20 | Practice | Positive + Biomedical | Negative + Biopsychosocial |
| 4 | 40 | Test | Positive + Biomedical | Negative + Biopsychosocial |
| 5 | 20 | Practice | Biomedical | Biopsychosocial |
| 6 | 20 | Practice | Positive + Biopsychosocial | Negative + Biomedical |
| 7 | 40 | Test | Positive + Biopsychosocial | Negative + Biomedical |

*For half of the participants, the pairings in blocks 3, 4, 6, and 7 were switched.

†For half of subjects, the handedness of both concepts and attributes were switched for all blocks.

permutations of an IAT based on the order and handedness (left vs. right) and surgeons were randomized into one of these 4 groups.

## Measured variables

Our primary outcome of interest was the d-score of the IAT, which is calculated based on the time it takes an observer to associate certain concepts, and ranges from -1.5 to 1.5. A d-score of 0 indicates no bias in either direction while a d-score of greater magnitude indicates greater implicit bias towards one habit of thinking. In this context, a more negative d-score indicates a stronger bias towards the biomedical paradigm. In the questionnaire portion of the survey, surgeons were asked four questions to gauge explicit preference for either the biomedical or biopsychosocial paradigm, rated on an 11-point ordinal scale (Table 4). Surgeons' explicit preference was converted to a total score on a scale from -50 to 50, with more positive scores indicating a stronger explicit preference for the biopsychosocial paradigm. We converted the 11-point ordinal scale to a scale ranging from -50 to +50 to make the presentation of the results more intuitive. For instance, whole numbers rather than decimals.

## Statistical analysis

Descriptive statistics were performed of all participants. Discrete variables were reported as number (percentage); continuous variables as mean ± standard deviation or median (interquartile range). Mean was used when the distribution of the variable was normal while median was used when the variable was non-normal or had significant skew. We did not do an a priori

**Table 4. Explicit bias survey questions.**

| Question* | Text |
|---|---|
| 1 | In my practice, I pay attention to symptoms of depression and other psychological factors that may influence symptom severity. |
| 2 | I am interested in scientific research that addresses the impact of mood, coping, and misconceptions about illness on the severity of symptoms that patients experience. |
| 3 | The majority of the variation in pain and functional limitations in orthopedic patients can be explained by variation in pathology severity, such as the grade of osteoarthritis. |
| 4 | Low mood and the presence of anxiety symptoms are often secondary to orthopedic illness and will improve after surgery. |

*All answers given on a scale from 0 (strongly disagree) to 10 (strongly agree).

power analysis because we had a fixed pool of respondents to draw from. Shapiro-Wilk tests were used to assess normality. Explicit bias scores were normally distributed while implicit bias scores were not. Student t-tests were used when analyzing relationships between explicit bias scores and binary variables. One-way analysis of variance (ANOVA) tests were used to analyze relationships between explicit bias scores and non-binary variables. Wilcoxon rank sum tests were used when analyzing relationships between implicit bias scores and binary variables. Kruskal-Wallis H tests were used for analyzing the relationships between implicit bias scores and non-binary variables. In multivariable analysis, the response variable was explicit preference for the biopsychosocial paradigm. The explanatory variables were practice location (continent) and orthopedic subspecialty. All explanatory variables with $P < 0.10$ in bivariate analysis were moved to multivariable linear regression analysis [34]. A single regression model was created with qualifying variables (practice location and subspecialty). Regression coefficients (RC), 95% Confidence Intervals (95% CI), standard errors, $P$-values, partial r-squared, and adjusted r-squared values were reported.

## Results

*Questions 1 and 2: What is the average relative implicit bias of musculoskeletal surgeons towards health as biomedical or biopsychosocial? What is their relative explicit preference?*

On average, surgeons reported an explicit preference for the biopsychosocial model (mean: 14; standard deviation: 14) (Fig 1) but had an implicit bias favoring the biomedical paradigm (d-score: -0.21; interquartile range [IQR]: -0.56 to 0.19) (Fig 2).

*Question 3: Which surgeon factors are associated with greater explicit preference or implicit bias towards either the biomedical or biopsychosocial paradigms?*

In bivariate analysis, men had a greater implicit bias favoring the biomedical paradigm (d-score: -0.30; IQR: -0.57 to 0.14) than women (d-score: 0.12; IQR: -0.23 to 0.65) ($P = 0.005$; Table 5). There were no other surgeon factors associated with implicit bias, and no multivariable analysis was performed.

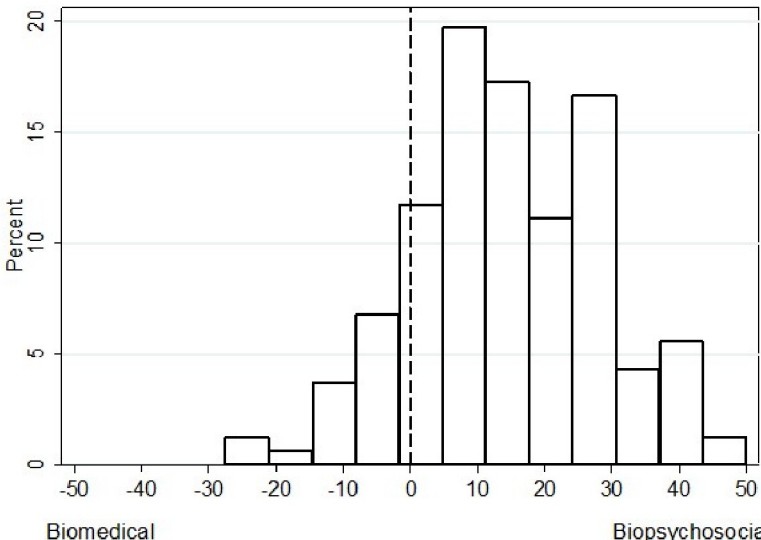

**Fig 1. Surgeons' explicit preference for the biomedical or biopsychosocial paradigm, scored on a scale from -50 to 50.**

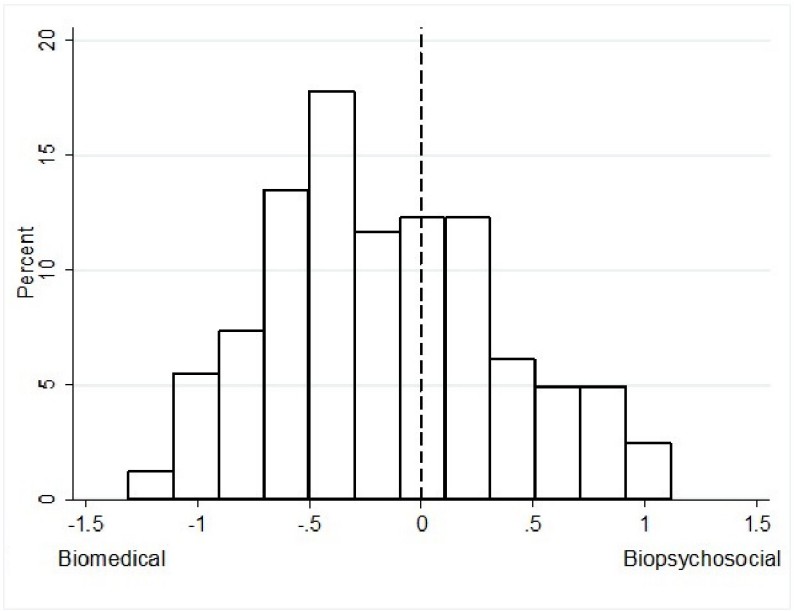

**Fig 2. Surgeons' implicit preference for the biomedical or biopsychosocial paradigm, scored on a scale from -1.5 to 1.5.**

**Table 5. Bivariate analysis of surgeon factors associated with the implicit and explicit bias for the biomedical or biopsychosocial paradigm.**

| Variable | Explicit bias | | Implicit bias | |
|---|---|---|---|---|
| | −50 = biomedical; 50 = biopsychosocial | | −1.5 = biomedical; 1.5 = biopsychosocial | |
| | Value (−50 to 50) | *P*-value | Value (−1.5 to 1.5) | *P*-value |
| Gender | | 0.393 | | **0.005** |
| Men | 15 ± 14 | | −0.30 (−0.57 to 0.14) | |
| Women | 18 ± 11 | | 0.12 (−0.23 to 0.65) | |
| Practice location | | **<0.001** | | 0.602 |
| United States | 20 ± 14 | | −0.23 (−0.53 to 0.15) | |
| Europe | 8.6 ± 11 | | −0.31 (−0.58 to 0.20) | |
| Other | 6.8 ± 12 | | −0.13 (−0.44 to 0.21) | |
| Years in practice | | 0.909 | | 0.419 |
| 0 to 5 | 14 ± 13 | | −0.033 (−0.45 to 0.23) | |
| 6 to 10 | 16 ± 13 | | −0.24 (−0.53 to 0.15) | |
| 11 to 20 | 15 ± 14 | | −0.38 (−0.62 to 0.14) | |
| More than 20 | 15 ± 15 | | −0.26 (−0.52 to 0.12) | |
| Subspecialty | | **<0.001** | | 0.930 |
| Hand and wrist | 21 ± 13 | | −0.38 (−0.57 to 0.20) | |
| Trauma | 10 ± 13 | | −0.16 (−0.57 to 0.062) | |
| Shoulder and elbow | 13 ± 11 | | −0.33 (−0.46 to 0.14) | |
| General orthopedics/Other | 15 ± 16 | | −0.15 (−0.53 to 0.20) | |
| Supervising trainees | | 0.106 | | 0.599 |
| Yes | 14 ± 13 | | −0.25 (−0.56 to 0.15) | |
| No | 19 ± 15 | | −0.24 (−0.54 to 0.21) | |

Continuous variables as mean ± standard deviation or median (interquartile range). All variables with *P* < **0.10** were moved to multivariable analysis.

**Table 6. Multivariable linear regression analysis of surgeon factors associated with the explicit preference for the biopsychosocial paradigm.**

| Variable | Regression Coefficient (95% Confidence Interval) | Standard error | P-value | Partial R$^2$ | Adjusted R$^2$ |
|---|---|---|---|---|---|
| | | | | | 0.16 |
| Practice location | | | | | |
| United States | *reference value* | | | | |
| Europe | -9.1 (-14 to -4.4) | 2.4 | **<0.001** | 0.089 | |
| Other | -6.5 (-13 to 0.32) | 3.5 | 0.062 | 0.023 | |
| Subspecialty | | | | | |
| Hand and wrist | *reference value* | | | | |
| Trauma | -6.2 (-11 to -1.0) | 2.6 | **0.019** | 0.036 | |
| Shoulder and elbow | -5.4 (-13 to 1.7) | 3.6 | 0.133 | 0.015 | |
| General orthopedics/Other | -3.2 (-9.3 to 2.9) | 3.1 | 0.299 | 0.0072 | |

**Bold** indicates statistical significance, $P < 0.05$.

Accounting for potential confounders in multivariable analysis, a greater explicit preference towards the biomedical paradigm was independently associated with a European practice location (Regression coefficient: -9.1; 95% CI: -14 to -4.4; $P$ <0.001), and trauma as a subspecialty (RC: -6.2; 95% CI: -11 to -1.0; $P$ <0.001) (Table 6).

*Question 4: Is there a correlation between the explicit preference and the implicit bias towards either paradigm?*

There was no correlation between the extent of explicit preference for, or implicit bias towards either paradigm (ρ = 0.10; $P$ = 0.19; Fig 3).

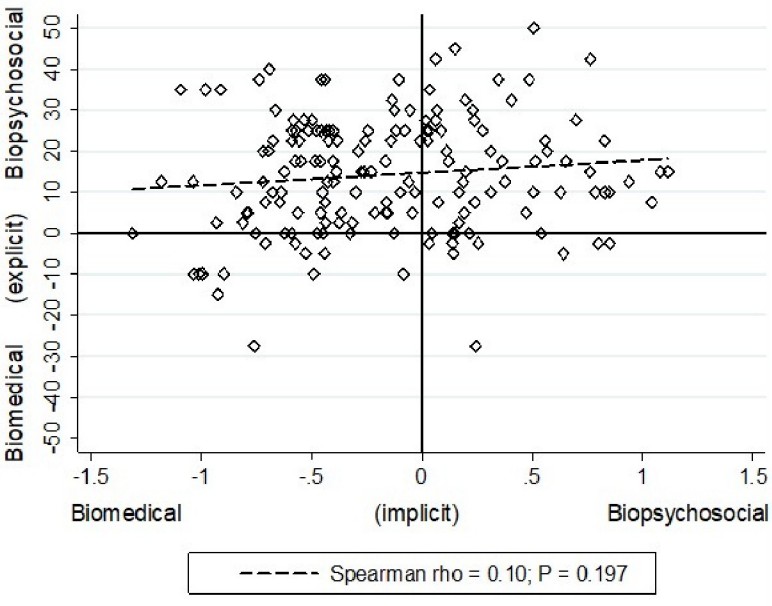

**Fig 3. Scatter plot of the implicit and explicit surgeon preferences for the biomedical or biopsychosocial paradigm.**

## Discussion

The notable variability in illness severity for a given pathology cannot be fully explained in the biomedical paradigm of human illness [12, 16]. Implementation of the evidence that mental and social factors have a notable association with symptom intensity [12, 14, 17, 35, 36] into daily practice might be hindered by habits of thought and action (unconscious bias) favoring the biomedical relative to the biopsychosocial model. We conducted a survey-based experiment of members of the Science of Variation Group and found that musculoskeletal surgeons reported an explicit preference for the biopsychosocial paradigm and had an implicit bias favoring the biomedical paradigm.

This study had several limitations. First, some words used in the IAT (e.g., therapy) are somewhat ambiguous, and may be interpreted variably (e.g., 'physical therapy' rather than 'mental health counseling'). In addition, our participants are an international group of surgeons, some of whom speak English as a second language. The training rounds support participants with the intended meaning and associations of these words, which should mitigate the effects of English dialect and proficiency. Second, the explicit preference was measured after the IAT, which could skew the explicit preference. However, there is evidence that the order has relatively little influence on the results [31]. Finally, the SOVG largely consists of white men in academic practice. The comparison of explicit preference between men and women, and between surgeons who do or do not supervise trainees could be relatively underpowered. However, we believe that differences between these individual subgroups are less generalizable and arguably less important than the finding of discordance between explicit preference and implicit bias towards the biomedical model. Additionally, the SOVG may tend to attract participation of surgeons who are interested in the subject-matter at hand, and this may constitute a form of selection bias. The observation of a relative implicit bias that health is biomedical, contrary to the expressed preference for the biopsychosocial paradigm, is notable in the subpopulation of the SOVG that chose to participate. The relative implicit bias could even be greater among other surgeons. Importantly, the part of the experiment measuring associations relies on within-sample variation to find associations. These findings might be relatively reproducible in samples with different mean implicit and explicit biases if there is sufficient within sample variation.

The finding that, on average, surgeons have an implicit relative biomedical bias and an explicit relative biopsychosocial preference demonstrates the ongoing transition towards whole-person models of care that anticipates the influence of mood, coping strategies, and hindrance of life roles on illness severity and recovery trajectories [17, 37]. In other words, surgeons understand the utility of the biopsychosocial model but are still working to evolve habits of thinking that unconsciously favor the biomedical model. The shift over the last decade from clinician-reported measures (for example, the Mayo Elbow Performance Index [38] and the Musculoskeletal Tumor Society MSTS scoring system [39]) to patient-rated measures of capability (e.g., Disabilities of the Arm, Shoulder and Hand [DASH] questionnaire [40] and the Patient-Reported Outcomes Measurement Information System [PROMIS] Physical Function) [41] illustrates the increasing interest in gauging health from the patient's perspective.

A variety of evidence-based strategies exist to mitigate bias [27]. Clinicians can acknowledge and affirm uncomfortable thoughts rather than suppressing them [42]. They can be conscientious of their gut feelings during certain situations. After recognizing uncomfortable thoughts and gut feelings, clinicians can consider changing settings and situations that may give rise to negative or stereotypical responses [43]. These may seem like "common sense" interventions, but they are difficult to implement in real-time, especially in high-pressure clinical situations. Specific training from experts may ease the implementation of these strategies,

particularly during malleable periods of training, such as residency or medical school. Individual-level interventions can be paired with system-level interventions to achieve meaningful changes. Policymakers can promote the biopsychosocial model by affirming it as part of core organizational and institutional values [44]. Additionally, policymakers and leaders can implement systems that promote the anticipation and detection of bias. They can respond to bias detection by changing institutions that propagate those biases.

The observations that a surgeon factor (gender) was associated with the degree of implicit bias, and practice location and subspecialty were associated with the degree of explicit preference, demonstrates the influence of varied experience, and has implications about the ability to cultivate new automatic thoughts about the biopsychosocial paradigm. Bias training, when performed effectively and administered to the correct target groups, can effectively improve the existence of harmful implicit and explicit biases [45–47]. Our findings that bias against the biopsychosocial paradigm may be associated with certain subgroups within the orthopedic surgery community may allow for targeting of those subgroups for effective elimination of those biases. There is little to gain from providing bias elimination training to those who do not harbor that bias [47].

The observation that there was no correlation between surgeon explicit preference and implicit bias for the biopsychosocial model is in line with prior research, suggesting that explicit preference and implicit bias are subject to different cognitive forces [48, 49]. Given that most of us received medical training embedded in the biomedical paradigm and given that this is still the paradigm in which most musculoskeletal care is designed and delivered, this mismatch between explicit preference and implicit bias is expected. The mismatch may also be due to the social desirability bias [50, 51] given that all surgeons in the database are familiar with the research by the senior authors, and it is possible that they gave socially desirable answers when asked directly about their interest in biopsychosocial research.

We performed an implicit association test among an international group of surgeons and found that most have an explicit relative preference for the biopsychosocial model and an implicit relative bias in favor of the biomedical paradigm. Implicit association testing has the potential to enhance surgeon self-awareness, professional and personal development, and quality improvement efforts. Greater surgeon awareness of implicit biases can aid the transition towards comprehensive care models that acknowledge the influence of environmental, mental, and social factors on illness severity. Future studies may measure the impact of biases towards the biomedical paradigm on surgeon participation in comprehensive care models.

## Supporting information

**S1 Dataset.**
(XLS)

## Author Contributions

**Conceptualization:** David Ring, Prakash Jayakumar.

**Data curation:** Sina Ramtin.

**Formal analysis:** Sina Ramtin, Tom Crijns.

**Methodology:** Sina Ramtin, Tom Crijns.

**Project administration:** David Ring, Prakash Jayakumar.

**Supervision:** David Ring.

**Writing – original draft:** Sina Ramtin, Dayal Rajagopalan, David Ring.

**Writing – review & editing:** Sina Ramtin, Dayal Rajagopalan, David Ring, Tom Crijns, Prakash Jayakumar.

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
