## [Decision Letter · Decision Letter 0]

10 May 2024

PONE-D-24-08426Musculoskeletal Surgeons Have Implicit Bias Towards the Biomedical Paradigm of Human IllnessPLOS ONE

Dear Dr. Ring,

Thank you for submitting your manuscript to PLOS ONE. After careful consideration, we feel that it has merit but does not fully meet PLOS ONE’s publication criteria as it currently stands. Therefore, we invite you to submit a revised version of the manuscript that addresses the points raised during the review process.

We look forward to receiving your revised manuscript.

Kind regards,

Adedayo Ajidahun

Academic Editor

PLOS ONE

Journal Requirements:

3. In the online submission form, you indicated that The data underlying the results presented in the study are available from Dr. David Ring.

Reviewers' comments:

Reviewer's Responses to Questions

**Comments to the Author**

1. Is the manuscript technically sound, and do the data support the conclusions?

Reviewer #1: Yes

Reviewer #2: Yes

2. Has the statistical analysis been performed appropriately and rigorously? 

Reviewer #1: Yes

Reviewer #2: Yes

3. Have the authors made all data underlying the findings in their manuscript fully available?

Reviewer #1: Yes

Reviewer #2: Yes

4. Is the manuscript presented in an intelligible fashion and written in standard English?

Reviewer #1: Yes

Reviewer #2: Yes

5. Review Comments to the Author

**Reviewer #1:** Methods: Line 117 - consider removing "We welcome new members and collaborators"

Is there any reason why the relationship between years of experience and implicit bias or explicit preference was not explored?

Authors should provide the test-retest reliability and internal consistency values of the IAT

**Reviewer #2:** This is a well written manuscript aside minor typo which is negligible but still need attention.

Few adjustments must be adhered to which has been highlighted in the corrected manuscript.

All the best with the final submission

6. PLOS authors have the option to publish the peer review history of their article (what does this mean?). If published, this will include your full peer review and any attached files.

Reviewer #1: **Yes: **Oluwagbemiga DadeMatthews

Reviewer #2: **Yes: **Olatunbosun Oriyomi Olaleye

---

## [Author Response · Author response to Decision Letter 0]

3 Jun 2024

Dear Editor and Reviewers. 

The file was uploaded. 

Thanks.

---

## [Decision Letter · Decision Letter 1]

26 Aug 2024

Musculoskeletal Surgeons Have Implicit Bias Towards the Biomedical Paradigm of Human Illness

PONE-D-24-08426R1

Dear Dr. Ring,

We’re pleased to inform you that your manuscript has been judged scientifically suitable for publication and will be formally accepted for publication once it meets all outstanding technical requirements.

Kind regards,

Adedayo Ajidahun

Academic Editor

PLOS ONE

Additional Editor Comments (optional):

Reviewers' comments:

Reviewer's Responses to Questions

**Comments to the Author**

1. If the authors have adequately addressed your comments raised in a previous round of review and you feel that this manuscript is now acceptable for publication, you may indicate that here to bypass the “Comments to the Author” section, enter your conflict of interest statement in the “Confidential to Editor” section, and submit your "Accept" recommendation.

Reviewer #1: All comments have been addressed

Reviewer #2: All comments have been addressed

2. Is the manuscript technically sound, and do the data support the conclusions?

Reviewer #1: Yes

Reviewer #2: Yes

3. Has the statistical analysis been performed appropriately and rigorously? 

Reviewer #1: Yes

Reviewer #2: Yes

4. Have the authors made all data underlying the findings in their manuscript fully available?

Reviewer #1: Yes

Reviewer #2: Yes

5. Is the manuscript presented in an intelligible fashion and written in standard English?

Reviewer #1: Yes

Reviewer #2: Yes

6. Review Comments to the Author

Reviewer #1: Thank you for your contribution to the knowledge of the explicit preference for the biopsychosocial model and implicit bias for the biomedical model in surgical care.

Reviewer #2: I will expect the author(s) to address all necessary comments and recommend prior to final submission

7. PLOS authors have the option to publish the peer review history of their article (what does this mean?). If published, this will include your full peer review and any attached files.

Reviewer #1: **Yes: **Oluwagbemiga DadeMatthews

Reviewer #2: No

---

## [Editor Report · Acceptance letter]

28 Aug 2024

PONE-D-24-08426R1 

PLOS ONE

Dear Dr. Ring, 

I'm pleased to inform you that your manuscript has been deemed suitable for publication in PLOS ONE. Congratulations! Your manuscript is now being handed over to our production team.

Kind regards, 

on behalf of

Dr. Adedayo Ajidahun 

Academic Editor

PLOS ONE